# Optimizer Amalgamation

**Tianshu Huang[1,2], Tianlong Chen[1], Sijia Liu[3], Shiyu Chang[4], Lisa Amini[5], Zhangyang Wang[1]**
[1]University of Texas at Austin, [2]Carnegie Mellon University, [3]Michigan State University,
[4]University of California, Santa Barbara, [5]MIT-IBM Watson AI Lab, IBM Research
`tianshu@cmu.edu`, {`tianlong.chen,atlaswang`}`@utexas.edu`,
`liusiji5@msu.edu`, `chang87@ucsb.edu`, `lisa.amini@ibm.com`

## Abstract

Selecting an appropriate optimizer for a given problem is of major interest for researchers and practitioners. Many analytical optimizers have been proposed using a variety of theoretical and empirical approaches; however, none can offer a universal advantage over other competitive optimizers. We are thus motivated to study a new problem named *Optimizer Amalgamation*: how can we best combine a pool of "teacher" optimizers into a single "student" optimizer that can have stronger problem-specific performance? In this paper, we draw inspiration from the field of "learning to optimize" to use a learnable amalgamation target. First, we define three differentiable amalgamation mechanisms to amalgamate a pool of analytical optimizers by gradient descent. Then, in order to reduce variance of the amalgamation process, we also explore methods to stabilize the amalgamation process by perturbing the amalgamation target. Finally, we present experiments showing the superiority of our amalgamated optimizer compared to its amalgamated components and learning to optimize baselines, and the efficacy of our variance reducing perturbations. Our code and pre-trained models are publicly available at http://github.com/VITA-Group/OptimizerAmalgamation.

## 1 Introduction

Gradient-based optimization is ubiquitous in machine learning; accordingly, a cottage industry of gradient-based optimizer design has emerged (Schmidt et al., 2020). These optimizers generally propose algorithms that aim to make the "best" parameter update for a computed gradient (Kingma & Ba, 2017; Liu et al., 2020), with some also modifying the location where the parameters are computed (Zhang et al., 2019b). However, each gradient-based optimizer claim specific problems where they hold performance advantages, none can claim to be universally superior. Due to the "No Free Lunch" theorem for optimization (Wolpert & Macready, 1997), no optimizer can provide better performance on a class of problems without somehow integrating problem-specific knowledge from that class.

Furthermore, problems such as training neural networks are not homogeneous. In the spatial dimension, different layers or even parameters can have different behavior (Chen et al., 2020b). Also, as evidenced by the popularity of learning rate schedules, neural network optimization also behaves very differently in the temporal dimension as well (Golatkar et al., 2019). This implies that no optimizer can provide the best performance for all parameters on a single problem or best performance over the entire optimization process.

In order to build a stronger optimizer, we propose the new problem of *optimizer amalgamation*: how can we best combine a pool of multiple "teacher" optimizers, each of which might be good in certain cases, into a single stronger "student" optimizer that integrates their strengths and offsets their weaknesses? Specifically, we wish for our combined optimizer to be adaptive both per-parameter and per-iteration, and exploit problem-specific knowledge to improve performance on a class of problems.

To "amalgamate" an optimizer from a pool of optimizers, we draw inspiration from recent work in *Learning to Optimize* (Chen et al., 2021a) which provides a natural way to parameterize and train optimization update rules. In Learning to Optimize, optimizers are treated as policies to be learned from data. These "learned" optimizers are typically parameterized by a recurrent neural network

---

[*]Work done while the author was at the University of Texas at Austin.

(Andrychowicz et al., 2016; Lv et al., 2017); then, the optimizer is meta-trained to minimize the loss of training problems, or "optimizees", by gradient descent using truncated back-propagation through time. Yet to our best knowledge, no existing work has leveraged those learnable parameterizations to amalgamate and combine analytical optimizers.

For our proposed formulation of optimizer amalgamation, we treat the learned optimizer as the amalgamation target. Then, we define amalgamation losses which can be used to combine feedback from multiple analytical optimizers into a single amalgamated optimizer, and present several amalgamation schemes. Finally, we explore smoothing methods that can be used during the amalgamation process to reduce the variance of the amalgamated optimizers. Our contributions are outlined below:

- We formulate the new problem of optimizer amalgamation, which we define as finding a way to best amalgamate a pool of multiple analytical optimizers to produce a single stronger optimizer. We present three schemes of optimizer amalgamation: additive amalgamation, min-max amalgamation, and imitation of a trained choice.
- We observe instability during the amalgamation process which leads to amalgamated optimizers having varied performance across multiple replicates. To mitigate this problem, we explore ways to reduce amalgamation variance by improving smoothness of the parameter space. We propose smoothing both by random noise or adversarial noise.
- We present experiments showing extensive and consistent results that validate the effectiveness of our proposal. Specifically, we find that more advanced amalgamation techniques and weight space training noise lead better average case performance and reduced variance. We also show that our amalgamation method performs significantly better than previous methods on all problems, with few exceptions.

## 2 RELATED WORKS

**Knowledge Distillation and Amalgamation**  The prototype of knowledge distillation was first introduced by (Bucilua et al., 2006), which used it for model compression in order train neural networks ("students") to imitate the output of more complex models ("teachers"). Knowledge distillation was later formalized by (Hinton et al., 2015), who added a temperature parameter to soften the teacher predictions and found significant performance gains as a result.

The success of knowledge distillation spurred significant efforts to explain its effectiveness. Notably, Chen et al. (2020c); Yuan et al. (2020) discovered that trained distillation teachers could be replaced by hand-crafted distributions. (Yuan et al., 2020) provided further theoretical and empirical explanation for this behavior by explicitly connecting Knowledge distillation to label smoothing, and (Ma et al.; Chen et al., 2021b) further credited the benefits of knowledge distillation to the improved *smoothness* of loss surfaces, which has been demonstrated to help adversarial training Cohen et al. (2019); Lecuyer et al. (2019) and the training of sparse neural networks Ma et al..

The potential of knowledge distillation to improve the training of neural networks also spurred diverse works extending knowledge distillation. For example, (Romero et al., 2015; Wang et al., 2018; Shen et al., 2018; 2019b; Ye et al., 2020b) propose using intermediate feature representations as distillation targets instead of just network outputs, and (Tarvainen & Valpola, 2017; Yang et al., 2018; Zhang et al., 2019a) unify student and teacher network training to reduce computational costs. Knowledge distillation has also been extended to distilling multiple teachers, which is termed Knowledge Amalgamation (Shen et al., 2019a; Luo et al., 2019; Ye et al., 2019; 2020a).

Although using output logits from pre-trained networks has been extensively explored in knowledge distillation, we study a new direction of research distilling optimization knowledge from sophisticated analytical optimizers to produce stronger "learned" optimizers, hence the name "optimizer amalgamation". Not only this is a new topic never studied by existing knowledge distillation literature, but also it needs to distill longitudinal output dynamics — not one final output — from multiple teachers.

**Learning to optimize**  Learning to Optimize is a branch of meta learning which proposes to replace hand-crafted analytical optimizers with *learned optimizers* trained by solving optimization problems, or *optimizees*. The concept was first introduced by (Andrychowicz et al., 2016), who used a Long Short-Term Memory (LSTM) based model in order to parameterize gradient-based optimizers. This model took the loss gradient as its input and output a learned update rule which was then trained by

gradient descent using truncated backpropagation through time. (Andrychowicz et al., 2016) also established a coordinate-wise design pattern, where the same LSTM weights are applied to each parameter of the optimizee in order to facilitate generalization to models with different architectures.

Building on this architecture, Wichrowska et al. (2017) and Lv et al. (2017) proposed improvements such as hierarchical architectures connecting parameter RNNs together and augmenting the gradient with additional inputs. Many methods have also been proposed to improve the training of learned optimizers such as random scaling and convex augmentation (Lv et al., 2017), curriculum learning and imitation learning (Chen et al., 2020a), and Jacobian regularization (Li et al., 2020). Notably, Chen et al. (2020a) also proposed a method of imitation learning, which can be viewed as a way of distilling a single analytical optimizer into a learned parameterization.

Learning to Optimize has been extended to a variety of other problems such as graph convolutional networks (You et al., 2020), domain generalization (Chen et al., 2020b), noisy label training (Chen et al., 2020c), adversarial training (Jiang et al., 2018; Xiong & Hsieh, 2020), and mixmax optimization (Shen et al., 2021). Moving away from gradient-based optimization, black-box optimization has also been explored (Chen et al., 2017; Cao et al., 2019). For a comprehensive survey with benchmarks, readers may refer to Chen et al. (2021a).

**Perturbations and Robustness**    The optimization process is naturally subject to many possible sources of noise, such as the stochastic gradient noise Devolder et al. (2011); Gorbunov et al. (2020); Simsekli et al. (2019) which is often highly non-Gaussian and heavy-tail in practice; the random initialization and (often non-optimal) hyperparameter configuration; the different local minimum reached each time in non-convex optimization Jain & Kar (2017); and the limited numerical precision in implementations De Sa et al. (2017). The seen and unseen optimizees also constitute domain shifts in our case. In order for a consistent and reliable amalgamation process, the training needs to incorporate resistance to certain perturbations of the optimization process.

We draw inspiration from deep learning defense against various random or malicious perturbations. For example, stability training Zheng et al. (2016) stabilizes deep networks against small input distortions by regularizing the feature divergence caused by adding random Gaussian noises to the inputs. Adversarial robustness measures the ability of a neural network to defend against malicious perturbations of its inputs (Szegedy et al., 2013; Goodfellow et al., 2014). For that purpose, random smoothening (Lecuyer et al., 2019; Cohen et al., 2019) and adversarial training (Madry et al., 2017) have been found to increase model robustness with regard to random corruptions or worst-case perturbations; as well as against testing-time domain shifts Ganin et al. (2016). Recent work (He et al., 2019; Wu et al., 2020) extends input perturbations to weight perturbations that explicitly regularize the flatness of the weight loss landscape, forming a double-perturbation mechanism for both inputs and weights.

**Other Approaches**    The problem of how to better train machine learning models has many diverse approaches outside the Learning to Optimize paradigm that we draw from, and forms the broader AutoML problem Hutter et al. (2018) together with model selection algorithms. Our approach falls under meta-learning, which also includes learned initialization approaches such as MAML (Finn et al., 2017) and Reptile (Nichol et al., 2018). Other optimizer selection and tuning methods include hypergradient descent (Baydin et al., 2017) and bayesian hyperparameter optimization (Snoek et al., 2012). Similar to our knowledge amalgamation approach, algorithm portfolio methods (where many algorithms are available, and some subset is selected) have also been applied to several problem domains such as Linear Programming (Leyton-Brown et al., 2003) and SAT solvers (Xu et al., 2008).

## 3    OPTIMIZER AMALGAMATION

### 3.1    MOTIVATION

Optimizer selection and hyperparameter optimization is a difficult task even for experts. With a vast number of optimizers to choose from with varying performance dependent on the specific problem and data (Schmidt et al., 2020), most practitioners choose a reasonable default optimizer such as SGD or Adam and tune the learning rate to be "good enough" following some rule of thumb.

As a consequence of the No Free Lunch theorem (Wolpert & Macready, 1997), the best optimizer to use for each problem, weight tensor within each problem, or each parameter may be different.

In practice, different layers within a given neural network can benefit from differently tuned hyper-parameters, for example by meta-tuning learning rates by layer (Chen et al., 2020b).

Accordingly, we wish to train an optimizer which is sufficiently versatile and adaptive at different stages of training and even to each parameter individually. Many methods have been proposed to parameterize optimizers in learnable forms including coordinate-wise LSTMs Andrychowicz et al. (2016); Lv et al. (2017), recurrent neural networks with hierarchical architectures Wichrowska et al. (2017); Metz et al. (2019), and symbolically in terms of predefined blocks Bello et al. (2017). Due to its high expressiveness and relative ease of training, we will use the workhorse of LSTM-based RNNProp architecture described by Lv et al. (2017) as our amalgamation target; more details about this architecture can be found in Appendix C.1.

## 3.2 THE BASIC DISTILLATION MECHANISM

Knowledge distillation can be viewed as regularizing the training loss with a distillation loss that measures the distance between teacher and student predictions (Hinton et al., 2015). In order to distill a pool of teacher optimizers $\boldsymbol{T} = T_1, T_2, \ldots T_k$ into our target policy $P$ by truncated backpropagation (Appendix A: Algorithm 1), we start by defining a training loss and amalgamation loss.

**Meta Loss**  In the context of training optimizers, the training loss is described by the *meta loss*, which is a function of the optimizee problem loss at each step (Andrychowicz et al., 2016). Suppose we are training a policy $P$ with parameters $\phi$ on a problem $\mathcal{M} : \mathcal{X} \to \mathbb{R}$ whose output is a loss for each point in data domain $\mathcal{X}$. During each iteration during truncated backpropagation through time, $P$ is used to compute parameter updates for $\mathcal{M}$ to obtain a trajectory of optimizee parameters $\theta_1, \theta_2, \ldots \theta_N$ where for the $i$th data batch $\boldsymbol{x}_i$ and parameters $\theta_i$ at step $i$, i.e. $\theta_{i+1} = \theta_i - P(\nabla_{\theta_i} \mathcal{M}(\boldsymbol{x}_i, \theta_i))$.

For some weighting function $f_1, f_2, \ldots f_N$, the meta loss is $\mathcal{L}_{\text{meta}}(\boldsymbol{x}, \theta_i; \phi) = \sum_{i=1}^{N} f_i(\mathcal{M}(\boldsymbol{x}_i, \theta_i))$; specifically, we will use the scaled log meta loss $f_i(m) = \log(m) - \log(\mathcal{M}(\boldsymbol{x}_i, \theta_0))$, which can be interpreted as the "mean log improvement."

**Distillation Loss**  The distillation loss in knowledge distillation measures the distance between teacher predictions and student predictions. In training optimizers, this corresponds to the distance between the optimization trajectories generated by the teacher and student. Suppose we have optimizee parameter trajectories $\boldsymbol{\theta}_i = (\theta_i^{(P)}, \theta_i^{(T)})$ generated by the teacher and student, respectively. Then, our distillation loss $\mathcal{L}_T$ for teacher $T$ is given by the $l_2$ log-loss:

$$\mathcal{L}_T(\boldsymbol{x}, \boldsymbol{\theta_i}; \phi) = \frac{1}{N} \sum_{i=1}^{N} \log \left\| \theta_i^{(P)} - \theta_i^{(T)} \right\|_2^2. \tag{1}$$

While knowledge distillation generally refers to imitating a model and imitation learning imitating a policy, the optimizer in our case can be regarded as both a model and a policy. As such, our loss function is similar to the imitation loss mechanism used by Chen et al. (2020a), which can be thought of as a special case of optimizer amalgamation where only a single teacher is used.

## 3.3 AMALGAMATION OF MULTIPLE TEACHER OPTIMIZERS: THREE SCHEMES

Now, what if there are multiple teachers that we wish to amalgamate into a single policy? How to best combine different knowledge sources is a non-trivial question. We propose three mechanisms:

(1) *Mean Amalgamation*: adding distillation loss terms for each of the optimizers with constant equal weights.
(2) *Min-max Amalgamation*: using a min-max approach to combine loss terms for each of the optimizers, i.e., "the winner (worst) takes all".
(3) *Optimal Choice Amalgamation*: First training an intermediate policy to choose the best optimizer to apply at each step, then distilling from that "choice optimizer".

**Mean Amalgamation**  In order to amalgamate our pool of teachers $\boldsymbol{T} = T_1, \ldots T_{|\boldsymbol{T}|}$, we generate $|\boldsymbol{T}| + 1$ trajectories $\boldsymbol{\theta}_i = (\theta_i^{(P)}, \theta_i^{(T_1)} \ldots \theta_i^{(T_{|\boldsymbol{T}|})})$ and add distillation losses for each teacher:

$$\mathcal{L}_{\text{mean}}(\boldsymbol{x}; \boldsymbol{\theta}_i; \phi) = \mathcal{L}_{\text{meta}}(\boldsymbol{x}; \theta_i^{(P)}; \phi) + \alpha \frac{1}{N} \sum_{i=1}^{N} \frac{1}{|\boldsymbol{T}|} \sum_{i=1}^{|\boldsymbol{T}|} \log \left\| \theta_i^{(P)} - \theta_i^{(T_i)} \right\|_2. \tag{2}$$

If we view knowledge distillation as a regularizer which provides soft targets during training, mean amalgamation is the logical extension of this by simply adding multiple regularizers to training.

An interesting observation is: when multiple teachers diverge, mean amalgamation loss tends to encourage the optimizer to choose one of the teachers to follow, potentially discarding the influence of all other teachers. This may occur if one teacher is moving faster than another in the optimizee space, or if the teachers diverge in the direction of two different minima. As this choice is a local minimum with respect to the mean log amalgamation loss, the optimizer may "stick" to that teacher, even if it is not the best choice.

**Min-Max Amalgamation**    In order to address this stickiness, we propose a second method: min-max amalgamation, where, distillation losses are instead combined by taking the maximum distillation loss among all terms at each time step:

$$\mathcal{L}_{\text{min-max}}(\boldsymbol{x}; \theta_i; \phi) = \mathcal{L}_{\text{meta}}(\boldsymbol{x}; \theta_i^{(P)}; \phi) + \alpha \frac{1}{N} \sum_{i=1}^{N} \max_{T \in \boldsymbol{T}} \log \left\| \theta_i^{(P)} - \theta_i^{(T)} \right\|_2. \tag{3}$$

This results in a v-shaped loss landscape which encourages the amalgamation target to be between the trajectories generated by the teacher pool and prevents the optimizer from "sticking" to one of the teachers.

One weakness shared by both mean and min-max amalgamation is memory usage. Both require complete training trajectories for each teacher in the pool to be stored in memory, resulting in memory usage proportional to the number of teachers, which limits the number of teachers that we could amalgamate from in one pool.

Min-max amalgamation also does not fully solve the problem of diverging teachers. While min-max amalgamation does ensure that no teacher is ignored, it pushes the amalgamation target to the midpoint between the optimizee weights of the two teachers, which does not necessarily correspond to a good optimizee loss. In fact, when teachers diverge into multiple local minima, any solution which considers all teachers must necessarily push the learned optimizer against the gradient, while any solution which allows the learned optimizer to pick one side must discard a number of teachers.

**Optimal Choice Amalgamation**    To fully unlock the power of knowledge amalgamation, we propose to solve the teacher divergence problem by first training an intermediate amalgamation target. By using only one teacher for a final distillation step, we remove the possibility of multiple teachers diverging while also allowing us to use more teachers without a memory penalty.

For optimizer pool $\boldsymbol{T}$, we define an choice optimizer $C$ which produces choices $c_1, c_2, \ldots c_N$ of which optimizer in the pool to apply at each time step, producing updates $\theta_{i+1}^{(C)} = \theta_i^{(C)} - T_{c_i}(\boldsymbol{g}_i)$. The objective of the choice optimizer is to minimize the meta loss $\mathcal{L}_{\text{meta}}(C; \boldsymbol{x})$ with respect to these choices $c_{1:N}$. We parameterize the choice function $C$ as a small two-layer LSTM, and train it by gradient descent. The LSTM takes the outputs of each optimizer in the pool, the layer type, and time step as inputs; more details are provided in Appendix C.1. To make it easier to train $C$ by truncated back-propagation through time, we relax the choices $c_{1:N}$ to instead be soft choices $\boldsymbol{c}_i \in \mathbb{R}^{|\boldsymbol{T}|} : c_i \geq 0, ||\boldsymbol{c}_i||_1 = 1$, resulting in the policy $\theta_{i+1}^{(C)} = \theta_i^{(C)} - \sum_{j=1}^{|\boldsymbol{T}|} c_i^{(j)} T_j(\boldsymbol{g}_i)$. Now, we use $C$ as a teacher to produce our final loss:

$$\mathcal{L}_{\text{choice}} = \mathcal{L}_{\text{meta}}(\phi; \boldsymbol{x}) + \alpha \frac{1}{N} \sum_{i=1}^{n} \log \left\| \theta_i^{(P)} - \theta_i^{(C)} \right\|_2. \tag{4}$$

## 4    STABILITY-AWARE OPTIMIZER AMALGAMATION

### 4.1    MOTIVATION

Modern optimization, even analytical, is subject to various forms of noise. For example, stochastic first-order method are accompanied with gradient noise (Devolder et al., 2011; Gorbunov et al., 2020; Simsekli et al., 2019) which is often highly non-Gaussian and heavy-tail in practice. Any non-convex optimization could reach different local minimum when solving multiple times (Jain &

Kar, 2017). When training deep neural networks, thousands or even millions of optimization steps are typically run, and the final outcome can be impacted by the random initialization, (often non-optimal) hyperparameter configuration, and even hardware precision (De Sa et al., 2017). Hence, it is highly desirable for optimizers to be stable: across different problem instances, between multiple training runs for the same problem, and throughout each training run (Lv et al., 2017).

Meta-training optimizers tends to be unstable. During the amalgamation process, we encounter significant variance where identically trained replicates achieve varying performance on our evaluation problems; this mirrors problems with meta-stability encountered by Metz et al. (2019). While amalgamation variance can be mitigated in small-scale experiments by amalgamating many times and using the best one, that variance represents a significant obstacle to large-scale training (i.e. on many and larger problems) and deployment of amalgamated optimizers. Thus, besides the aforementioned optimization stability issues, we also need to consider *meta-stability*, denoting the relative performance of optimizers across meta-training replicates.

In order to provide additional stability to the amalgamation process, we turn to adding noise during training, which is known to improve smoothness (Chen & Hsieh, 2020; Lecuyer et al., 2019; Cohen et al., 2019) and in turn improve stability (Miyato et al., 2018). Note that one can inject either random noise or adversarial perturbations onto either the input or the weight of the learnable optimizer. While perturbing inputs is more common, recent work (Wu et al., 2020) identified that a flatter weight loss landscape (loss change with respect to weight) leads to smaller robust generalization gap in adversarial training, thanks to its more "global" worst-case view.

We also discover in our experiments that perturbing inputs would make the meta-training hard to converge, presumably because the inputs to optimizers (gradients, etc.) already contain large amounts of batch noise and do not tolerate further corruption. We hence focus on perturbing optimizer weights for smoothness and stability.

## 4.2 Weight Space Perturbation for Smoothness

Weight space smoothing produces a noised estimate of the loss $\tilde{\mathcal{L}}$ by adding noise to the optimizer parameters $\phi$. By replacing the loss $\mathcal{L}(\phi, \boldsymbol{x})$ with a noisy loss $\tilde{\mathcal{L}} = \mathcal{L}(\tilde{\phi}, \boldsymbol{x})$, we encourage the optimizer to be robust to perturbations of its weights, increasing the meta-stability. We explore two mechanisms to increase weight space smoothness during training, by adding (1) a *random perturbation* to the weights as a gradient estimator, and (2) an *adversarial perturbation* in the form of a projected gradient descent attack (PGD).

Though new to our application, these two mechanisms have been adopted for other problems where smoothness is important such as neural architecture search (Chen & Hsieh, 2020) and adversarial robustness (Lecuyer et al., 2019; Cohen et al., 2019).

**Random Gaussian Perturbation**   In the first type of noise, we add gaussian noise with variance $\sigma^2$ to each parameter of the optimizer at each iteration, $\tilde{\phi} = \phi + \mathcal{N}(0, \sigma^2 I)$.

Since optimizer weights tend to vary largely in magnitude especially between different weight tensors, we modify this gaussian noise to be adaptive to the magnitude of the $l_2$ norm of each weight tensor $\tilde{\phi}^{(w)}$. For tensor size $|\phi^{(w)}|$, the added noise is given by

$$\tilde{\phi}^{(w)} = \phi^{(w)} + \mathcal{N}\left(0, \sigma^2 \frac{||\phi^{(w)}||_2^2}{|\phi^{(w)}|} I\right). \tag{5}$$

**Projected Gradient Descent**   For the second type of noise, we use adversarial noise obtained by projected gradient descent (Appendix A, Algorithm 2). For $A$ adversarial steps, the noised parameters are given by $\tilde{\phi} = \phi + \psi_A$, where $\psi_0 = \boldsymbol{0}$, and $\psi_{i+1} = \psi_i + \eta \operatorname{clip}_\varepsilon(\nabla_{\tilde{\psi}_i} \mathcal{L})$ for optimizer loss $\mathcal{L}$.

As with random Gaussian perturbations, we also modify the adversarial perturbation to be adaptive with magnitude proportional to the $l_2$ norm of each weight tensor $\phi$. Here, the adversarial attack step for weight tensor $w$ is instead given by

$$\psi_{i+1}^{(w)} = \psi_i^{(w)} + \frac{\varepsilon ||\phi||_2 \nabla_{\psi_i^{(w)}} \mathcal{L}}{||\nabla_{\psi_i^{(w)}} \mathcal{L}||_2}. \tag{6}$$

## 5 EXPERIMENTS

**Optimizee Details** All optimizers were amalgamated using a 2-layer convolutional neural network (CNN) on the MNIST (LeCun & Cortes, 2010) dataset (shortened as "Train") using a batch size of 128. During evaluation, we test the generalization of the amalgamated optimizer to other problems: ❶ *Different Datasets*: FMNIST (Xiao et al., 2017) and SVHN (Netzer et al., 2011). We also run experiments on CIFAR (Krizhevsky et al., 2009); since the Train network is too small to obtain reasonable performance on CIFAR, we substitute it for the Wider architecture and a 28-layer ResNet (He et al., 2015) labelled "CIFAR" and "ResNet" respectively. ❷ *Different Architectures*: a 2-layer MLP (MLP), a CNN with twice the number of units in each layer (Wider), and a deeper CNN (Deeper) with 5 convolutional layers. ❸ *Training settings*: training with a smaller batch size of 32 (Small Batch). We also try a new setting of training with differential privacy (Abadi et al., 2016) (MNIST-DP). Appendix B provides full architecture and training specifications.

**Optimizer Pool** We use two different optimizer pools in our experiment: "small," which consists of Adam and RMSProp, and "large," which also contains SGD, Momentum, AddSign, and PowerSign. Each optimizer has a learning rate tuned by grid search over a grid of $\{5 \times 10^{-4}, 1 \times 10^{-3}, 2 \times 10^{-3}, \dots 1\}$. The selection criteria is the best validation loss after 5 epochs for the Train network on MNIST, which matches the meta-training settings of the amalgamated optimizer. Appendix C.2 describes the optimizers used and other hyperparameters.

**Baselines** First, we compare our amalgamated optimizer against our analytical optimizer teachers which are combined into a "oracle optimizer," which is the optimizer in our pool of teachers with the best validation loss. We also compare against the optimal choice optimizer used in amalgamation, which functions like a per-iteration trained approximation of the oracle optimizer. Then, we evaluate previous learned optimizer methods: the original "Learning to Learn by Gradient Descent by Gradient Descent" optimizer Andrychowicz et al. (2016) which we refer to as "Original", RNNProp (Lv et al., 2017), a hierarchical architecture presented by "Learned Optimizers that Scale and Generalize" (Wichrowska et al., 2017) which we refer to as "Scale", and the best setup from Chen et al. (2020a) which shorten as "Stronger Baselines."

**Training and Evaluation Details** The RNNProp amalgamation target was trained using truncated backpropagation though time with a constant truncation length of 100 steps and total unroll of up to 1000 steps and meta-optimized by Adam with a learning rate of $1 \times 10^{-3}$. For our training process, we also apply random scaling (Lv et al., 2017) and curriculum learning (Chen et al., 2020a); more details about amalgamation training are provided in Appendix C.3. Amalgamation takes up to 6.35 hours for optimal choice amalgamation using the large pool and up to 10.53 hours when using adversarial perturbations; a full report of training times is provided in Appendix C.4.

For each optimizer amalgamation configuration tested, we independently trained 8 replicate optimizers. Then, each replicate was evaluated 10 times on each evaluation problem, and trained to a depth of 25 epochs each time. Finally, we measure the stability of amalgamated optimizers by defining three notions of stability for meta-trained optimizers: ❶ *Optimization stability*: the stability of the optimizee during the optimization process. Viewing stability of the validation loss as a proxy for model stability with respect to the true data distribution, we measure the epoch-to-epoch variance of the validation loss after subtracting a smoothed validation loss curve (using a Gaussian filter). ❷ *Evaluation stability*: the variance of optimizer performance across multiple evaluations. We find that the evaluation stability is roughly the same for all optimizers (Appendix E.1). ❷ *Meta-stability*: the stability of the amalgamation process, i.e. the variance of amalgamation replicates after correcting for evaluation variance. Meta-stability and evaluation stability are jointly estimated using a linear mixed effects model. The stability is reported as a standard deviation. More details are in Appendix D.

### 5.1 OPTIMIZER AMALGAMATION

**Amalgamation Methods** Figure 1 compares the mean performance of the three amalgamation methods with the small pool and Choice amalgamation with the large pool. Mean and min-max amalgamation were not performed on the large pool due to memory constraints. The amalgamated optimizers using optimal choice amalgamation perform better than Mean and Min-Max amalgamation. The size of the optimizer pool does not appear to have a significant effect in Optimal Choice amalgamation, with small pool and large pool amalgamated optimizers obtaining similar results.

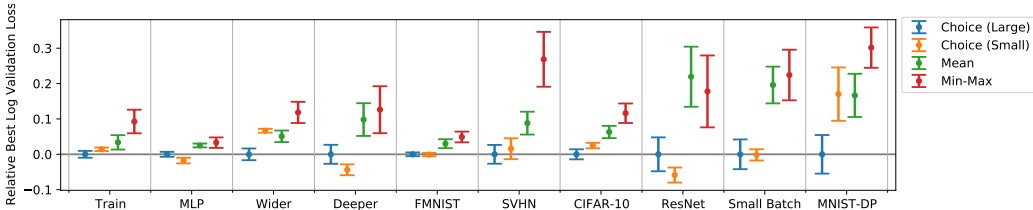

Figure 1: Amalgamated optimizer performance as measured by the best log validation loss and log training loss (lower is better) after 25 epochs; 95% confidence intervals are shown, and are estimated by a linear mixed effects model (Appendix D). In order to use a common y-axis, the validation loss is measured relative to the mean validation loss of the optimizer amalgamated from the large pool using optimal Choice amalgamation.

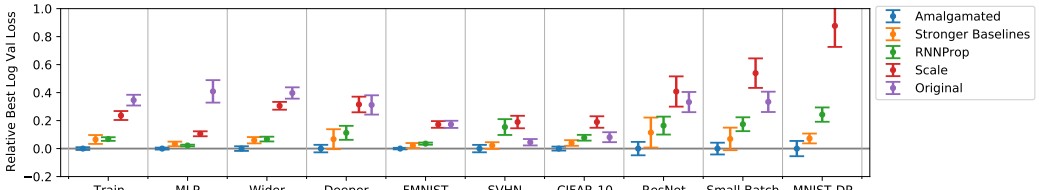

Figure 2: Comparison with other learned optimizers; for each problem, 95% confidence intervals of the mean are computed using a linear mixed effects model (Appendix D). Error bars are normalized by subtracting the mean log validation loss of the amalgamated optimizer to use the same Y-axis. Uncropped and accuracy versions can be found in Appendix E.3. The amalgamated optimizer performs better than other learned optimizers on all problems, and is significantly better except in some problems when compared to the Stronger Baselines trained RNNProp (Chen et al., 2020a).

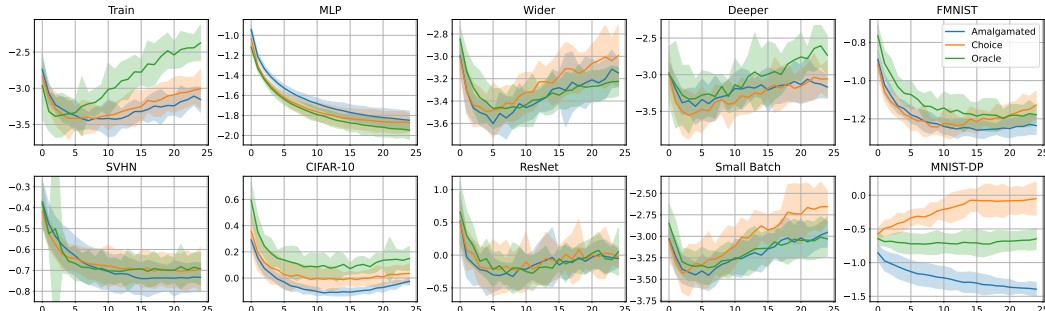

Figure 3: Comparison between the best Amalgamated Optimizer (blue), the optimal Choice optimizer used to train it (orange), and the oracle optimizer (grange); the shaded area shows ±2 standard deviations from the mean. The title of each plot corresponds to an optimizee; full definitions can be found in Appendix B. An version of this plot showing validation accuracy can be found in Appendix E.3. The amalgamated optimizer performs similarly or better than the choice optimizer and Oracle analytical optimizer on problems spanning a variety of training settings, architectures, and datasets.

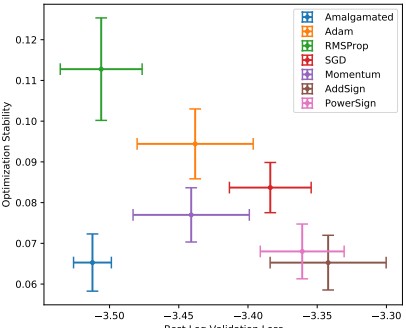

Figure 4: Relationship between optimization stability and performance as measured by validation loss on the Train optimizee; smaller stability and validation loss are better. Error bars show 95% confidence intervals; analytical optimizers in the large pool and the optimizer amalgamated from the large pool using Optimal Choice are shown.

**Previous Learned Optimizers**  Figure 2 compares the amalgamated optimizer against the baselines from Learning to Optimize. Optimizer amalgamation performs significantly better than all previous methods on all problems, with few exceptions (where it performs better but not significantly better).

**Analytical Optimizers**  In Figure 3, we compare the best replicate amalgamated from the large pool using Choice amalgamation with the "oracle optimizer" described above. The amalgamated optimizer achieves similar or better validation losses than the best analytical optimizers, indicating that our amalgamated optimizer indeed captures the "best" loss-minimization characteristics of each optimizer.

The amalgamated optimizer also benefits from excellent optimization stability, meeting or exceeding the optimization stability of the best analytical optimizers in the large

Figure 5: Optimization stability (lower is more stable) of an optimizer amalgamated by Optimal Choice from the large pool compared to optimization stability of the optimizers in that pool; 95% confidence intervals are shown. A larger version of this figure showing training loss and validation loss as well is provided in Appendix E.3.

pool (Figure 5). Comparing analytical optimizers, we observe a general inverse relationship between optimization performance and optimization stability: in order to achieve better optimization, an optimizer typically sacrifices some optimization stability in order to move faster through the optimizee weight space. By integrating problem-specific knowledge, the amalgamated optimizer is able to combine the best optimization performance and optimization stability characteristics (Figure 4).

## 5.2 STABILITY-AWARE OPTIMIZER AMALGAMATION

**Input Perturbation**  While we also tested perturbing the inputs of the optimizer during amalgamation, we were unable to improve stability. These experiments are included in Appendix E.4.

**Random Perturbation**  Min-max amalgamation was trained on the small optimizer pool with random perturbation relative magnitudes of $\varepsilon = \{5 \times 10^{-4}, 10^{-3}, 2 \times 10^{-3}, 5 \times 10^{-3}, 10^{-2}\}$. $\varepsilon = 10^{-1}$ was also tested, but all replicates tested diverged and are not reported here.

Comparing perturbed amalgamation against the non-perturbed baseline ($\varepsilon = 0$), we observe that perturbations increase meta-stability up to about $\varepsilon = 10^{-3}$ (Figure 6). For larger perturbation magnitudes, meta-stability begins to decrease as the perturbation magnitude overwhelms the weight "signal," eventually causing the training process to completely collapse for larger perturbation values. While the stability with random perturbation $\varepsilon = 10^{-2}$ is better than $10^{-3}$, this is likely due to random chance, since we use a small sample size of 8 replicates.

**Adversarial Perturbation**  Since adversarial perturbation is more computationally expensive than random perturbations, min-max amalgamation was tested on a coarser grid of relative magnitudes $\varepsilon = \{10^{-4}, 10^{-3}, 10^{-2}\}$, and to an adversarial attack depth of 1 step. These results are also reported in Figure 6, with $\varepsilon = 10^{-2}$ omitted since all replicates diverged during training.

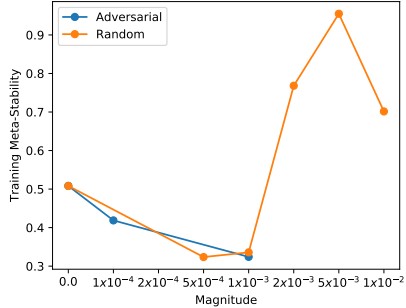

Figure 6: Amalgamation meta-stability for varying magnitudes of random and adversarial perturbations (lower is better). Meta-stability is measured by the variance across replicates of the training loss after 25 epochs on the Train convolutional network, adjusted for the variance of evaluation.

From our results, we observe that adversarial perturbations are about as effective as random perturbations. We also observe that the maximum perturbation magnitude that the amalgamation process can tolerate is much smaller for adversarial perturbations compared to random perturbations, likely because adversarial perturbations are much "stronger." Due to the significantly larger training cost of adversarial perturbations, we recommend random perturbations for future work.

**Application to Other Methods**  Random and Adversarial perturbations can be applied to any gradient-based optimizer meta-training method, including all of our baselines. An experiment applying Gaussian perturbations to the RNNProp baseline can be found in Appendix E.5.

## 6 CONCLUSION

We define the problem of optimizer amalgamation, which we hope can inspire better and faster optimizers for researchers and practitioners. In this paper, we provide a procedure for optimizer amalgamation, including differentiable optimizer amalgamation mechanisms and amalgamation stability techniques. Then, we evaluate our problem on different datasets, architectures, and training settings to benchmark the strengths and weaknesses of our amalgamated optimizer. In the future, we hope to bring improve the generalizability of amalgamated optimizers to even more distant problems.

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

# A ALGORITHMS

In this section, we provide a detailed description of the key algorithms used in our paper.

**Truncated Back-propagation**: Algorithm 1 shows truncated back-propagation applied to optimizer amalgamation. For an unrolling length $N$, $N$ data points (batches, in the case of mini-batch SGD) are sampled, which are split into $N/t$ truncations of length $t$. Note that this requires $N$ to be divisible by $t$; in our implementation, we require $t$ and $N/t$ to be specified as integers. For each truncation, the optimizee and teachers are trained for $t$ iterations, and meta-gradients are computed over that truncation and applied.

**Adversarial Weight Perturbation**: Algorithm 2 shows adversarial perturbations applied to optimizer amalgamation. For each adversarial attack step, meta-gradients are taken with respect to the parameters, and are normalized for each tensor with respect to its tensor norm before being applied as an adversarial perturbation.

---

**Algorithm 1:** Distillation by Truncated Back-propagation

**Inputs** : Amalgamation loss $\mathcal{L}_a$
Policy $P$ with parameters $\phi$
Teacher policies $\boldsymbol{T} = T_1, \ldots T_{|T|}$
Optimizee $\mathcal{M}, \boldsymbol{X}, \theta_0$
Unrolling, truncation lengths $N, t$
**Outputs** : Updated policy parameters $\phi$
Sample $N$ data points $\boldsymbol{x}_1, \ldots \boldsymbol{x}_N$ from $\boldsymbol{X}$.
$\theta_0^{(P)} = \theta_0^{(T_1)} = \ldots = \theta_0^{(T_{|T|})} = \theta_0$
**for** $i = 1, 2, \ldots N/t$ **do**
  **for** $j = 1, 2, \ldots t$ **do**
    $n = (i-1)t + j$
    Update optimizee for $P$:
    $\theta_{n+1}^{(P)} \leftarrow \theta_n^{(P)} - P\left[\nabla \mathcal{M}(\boldsymbol{x}_n, \theta_n^{(P)})\right]$
    Update optimizees for each teacher:
    **for** $k = 1, \ldots |\boldsymbol{T}|$ **do**
      $\theta_{n+1}^{(T_k)} \leftarrow$
      $\theta_n^{(T_k)} - T_k\left[\nabla \mathcal{M}(\boldsymbol{x}_n, \theta_n^{(T_k)})\right]$
    **end**
  **end**
  Compute distillation loss:
  $\mathcal{L}_i \leftarrow \mathcal{L}_a(\boldsymbol{x}_{[(i-1)t:it]}, \boldsymbol{\theta}_{[(i-1)t:it]}; \phi)$
  Update $\phi$ using $\nabla \mathcal{L}_i$
**end**

---

**Algorithm 2:** Adversarial Weight Perturbation for Truncated Back-propagation

**Inputs** : Truncated back-propagation
parameters $\mathcal{L}_a, P, \phi, \boldsymbol{T}, \mathcal{M}, \boldsymbol{X},$
$\theta_0, N, t$
Adversarial attack steps $A$
**Outputs** : Updated policy parameters $\phi$
Sample $N$ data points and initialize
optimizee parameters
**for** $i = 1, 2, \ldots N/t$ **do**
  $\psi_0 \leftarrow \boldsymbol{0}$
  **for** $a = 1, 2, \ldots A$ **do**
    Compute trajectories $\boldsymbol{\theta}_{[(i-1)t:it]}$ for
    $P$ and $\boldsymbol{T}$
    $\mathcal{L}_i^{(a)} \leftarrow \mathcal{L}_a($
      $\boldsymbol{x}_{[(i-1)t:it]}, \boldsymbol{\theta}_{[(i-1)t:it]}; \phi + \psi_a)$
    **for** *each weight tensor $w$* **do**
      $\gamma \leftarrow \dfrac{\nabla_{\psi_i^{(w)}} \mathcal{L}_i^{(a)}}{||\nabla_{\psi_i^{(w)}} \mathcal{L}_i^{(a)}||_2}$
      $\psi_a^{(w)} \leftarrow \psi_{a-1} + \varepsilon ||\phi||_2 \gamma.$
    **end**
  **end**
  $\mathcal{L}_i \leftarrow \mathcal{L}_a($
    $\boldsymbol{x}_{[(i-1)t:it]}, \boldsymbol{\theta}_{[(i-1)t:it]}; \phi + \psi_A)$
  Update $\phi$ using $\nabla \mathcal{L}_i$
**end**

---

# B OPTIMIZEE DETAILS

Table 1 shows a summary of the training problems used. While all training is performed on a 2-layer CNN on MNIST, we evaluated our optimizer on 4 different datasets described in (B.1) and 5 different architectures (described in B.2). We also experiment with different training settings, which are described in B.3.

## B.1 DATASETS

All datasets used are classification datasets, with cross entropy used as the training loss. The MNIST dataset (LeCun & Cortes, 2010) is used during training; the other datasets are, from most to least similar, are:

Table 1: Summary of Optimizee Specifications. Dataset, architecture, and training setting specifications are given in sections B.1, B.2, and B.3 respectively.

| Optimizee Name | Dataset | Architecture | Parameters | Other Settings |
|---|---|---|---|---|
| Train | MNIST | 2-layer CNN | 18122 | - |
| MLP | MNIST | 2-layer MLP | 15910 | - |
| Wider | MNIST | 2-layer CNN, 2x width | 61834 | - |
| Deeper | MNIST | 6-layer CNN | 72042 | - |
| FMNIST | FMNIST | 2-layer CNN | 18122 | - |
| SVHN | SVHN | 2-layer CNN | 21290 | - |
| CIFAR | CIFAR-10 | 2-layer CNN, 2x width | 68170 | - |
| ResNet | CIFAR-10 | 28-layer ResNet | 372330 | - |
| Small Batch | MNIST | 2-layer CNN | 18122 | Batch size 32 |
| MNIST-DP | MNIST | 2-layer CNN | 18122 | Differentially Private |

- FMNIST: Fashion MNIST (Xiao et al., 2017). FMNIST is also a drop-in replacement for MNIST with 10 classes and 28x28 grayscale images. Unlike MNIST or KMNIST, it features images of clothing instead of handwritten characters.
- SVHN: Street View House Numbers, cropped (Netzer et al., 2011). While SVHN also has 10 classes of numerical digits, the images are 32x32 RGB, and have significantly more noise than MNIST including "distraction digits" on each side.
- CIFAR-10 (Krizhevsky et al., 2009): the least similar dataset. While CIFAR-10 still has 10 classes and 32x32 RGB images, it has much higher noise and within-class diversity.

Sample images from these datasets are shown in Figure 7. All datasets were accessed using Tensor-Flow Datasets and have a CC-BY 4.0 license.

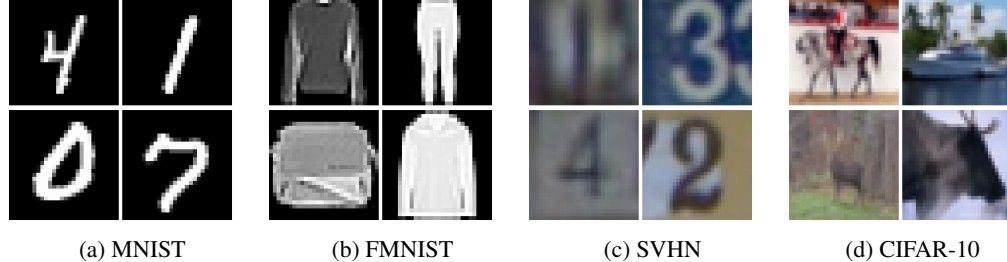

(a) MNIST          (b) FMNIST          (c) SVHN          (d) CIFAR-10

Figure 7: Dataset sample images.

## B.2    ARCHITECTURES

The Train convolutional network (Table 2a) has one convolution layer with 16 3x3 filters and one convolution layer with 32 5x5 filters. Each convolution layer uses ReLU activation, has stride 1x1, and is followed by a max pooling layer with size 2x2. Finally, a fully connected softmax layer is used at the output.

The four architectures evaluated are:

1. MLP: a 2-layer MLP with 20 hidden units and sigmoid activation
2. Wider: a modified version of Train with double width on each layer (Table 2b)
3. Deeper: a deeper network with 5 convolutional layers instead of 2 (Table 2c), again using ReLU activation and 1x1 stride
4. ResNet: a 28-layer ResNet (He et al., 2015) (Table 2d)

(a) Train

| Layer | Size | Units |
|---|---|---|
| Conv | 3x3 | 16 |
| Max Pool | 2x2 | - |
| Conv | 5x5 | 32 |
| Max Pool | 2x2 | - |
| Dense | - | 10 |

(b) Wider

| Layer | Size | Units |
|---|---|---|
| Conv | 3x3 | 32 |
| Max Pool | 2x2 | - |
| Conv | 5x5 | 64 |
| Max Pool | 2x2 | - |
| Dense | - | 10 |

(c) Deeper

| Layer | Size | Units |
|---|---|---|
| Conv | 3x3 | 16 |
| Conv | 3x3 | 32 |
| Conv | 3x3 | 32 |
| Max Pool | 2x2 | - |
| Conv | 3x3 | 64 |
| Conv | 3x3 | 64 |
| Max Pool | 2x2 | - |
| Dense | - | 10 |

(d) ResNet

| Block | Size | Units |
|---|---|---|
| Conv | 3x3 | 16 |
| Conv | 3x3 | 64 |
| Residual (4x) | 3x3 | 64 |
| Max Pool | 2x2 | - |
| Conv | 3x3 | 128 |
| Residual (4x) | 3x3 | 128 |
| Max Pool | 2x2 | - |
| Conv | 3x3 | 256 |
| Residual (4x) | 3x3 | 256 |
| Average Pool | Global | - |
| Dense | - | 10 |

Table 2: Convolutional Optimizee Architectures. Note that for the 28-layer ResNet, each residual block consists of 2 layers, adding up to 28 convolutional layers in total.

### B.3 OPTIMIZEE TRAINING

During training, a batch size of 128 is used except for the Small Batch evaluation, which has a batch size of 32. During training and evaluation, datasets are reshuffled each iteration.

To match the warmup process used in meta-training, warmup is also applied during evaluation. The SGD learning rate is fixed at 0.01, which is a very conservative learning rate which does not optimize quickly, but is largely guaranteed to avoid divergent behavior.

For differentially private training, we implement differentially private SGD (Abadi et al., 2016). In differentially private SGD, gradients are first clipped to a fixed $l_2$ norm $\varepsilon$ on a per-sample basis; then, gaussian noise with standard deviation $\sigma\varepsilon$ where $\sigma > 1$ is added to the aggregated batch gradients. In our experiments, we use clipping norm $\varepsilon = 1.0$ and noise ratio $\sigma = 1.1$. Both MNIST and KMNIST are used as training sets in order to simulate transfer from a non-private dataset (MNIST) used for meta-training to a private dataset (KMNIST).

## C AMALGAMATION DETAILS

### C.1 OPTIMIZER ARCHITECTURES

In this section, we provide the exact architecture specifications and hyperparameters of our amalgamated optimizer along with other training details and training time. Our implementation is open source, and can be found here: http://github.com/VITA-Group/OptimizerAmalgamation.

#### C.1.1 RNNPROP ARCHITECTURE

For our amalgamation target, we use RNNProp architecture described by Lv et al. (2017). For each parameter on each time step, this architecture takes as inputs RMSProp update $g/\sqrt{\tilde{v}}$ and Adam update $\hat{m}/\sqrt{\hat{v}}$ using momentum ($\hat{m}$) decay parameter $\beta_1 = 0.9$ and variance ($\hat{v}$) decay parameter $\beta_2 = 0.999$, matching the values used for our analytical optimizers. These values pass through a 2-layer LSTM with tanh activation, sigmoid recurrent activation, and 20 units per layer. The output of this 2-layer LSTM is passed through a final fully connected layer with tanh activation to produce a scalar final update for each parameter.

### C.1.2 CHOICE NETWORK ARCHITECTURE

Our Choice network for Optimal Choice Amalgamation is a modified RNNProp architecture. The update steps for each analytical optimizer are given as inputs to the same 2-layer LSTM used in RNNProp. Additionally, the current time step and tensor number of dimensions are provided, with the number of dimensions being encoded as a one-hot vector.

Then, instead of directly using the output of a fully connected layer as the update, LSTM output passes through a fully connected layer with one output per optimizer in the pool. This fully connected layer has a softmax activation, and is used as weights to combine the analytical optimizer updates.

### C.2 OPTIMIZER POOL

We consider six optimizers as teachers in this paper: Adam, RMSProp, SGD, Momentum, AddSign, and PowerSign. These optimizers are summarized in table 3.

Joining the popular hand-crafted optimizers Adam, RM-SProp, SGD, and Momentum, AddSign and PowerSign are two optimizers discovered by neural optimizer search (Bello et al., 2017). These two optimizers share the design principle that update steps should be larger when the momentum and gradient are in agreement:

$$\text{AddSign} \propto g(1 + \text{sign}(\hat{m})\text{sign}(g))$$
$$\text{PowerSign} \propto g\exp(\text{sign}(\hat{m})\text{sign}(g)). \tag{7}$$

Here, $g$ represents the gradient, $\hat{m}$ an exponential moving average of the gradient. In order to use AddSign

Table 3: Optimizer pool update rules; all updates include an additional learning rate hyperparameter.

| Optimizer | Update Rule |
|---|---|
| SGD | $g$ |
| Momentum | $\hat{m}$ |
| RMSProp | $g/\sqrt{\hat{v}}$ |
| Adam | $\hat{m}/\sqrt{\hat{v}}$ |
| AddSign | $g(1 + \text{sign}(\hat{m})\text{sign}(g))$ |
| PowerSign | $g\exp(\text{sign}(\hat{m})\text{sign}(g))$ |

and Powersign as teachers for gradient-based distillation, we modify them to be differentiable by replacing the sign function with a scaled tanh with magnitudes normalized by $\sqrt{\hat{v}}$:

$$\text{sign}(\hat{m})\text{sign}(g) \approx \tanh(\hat{m}/\sqrt{\hat{v}})\tanh(g/\sqrt{\hat{v}}) \tag{8}$$

By dividing by $\sqrt{\hat{v}}$, we provide a consistent magnitude to the tanh function so that sign agreement mechanism is not affected by overall gradient magnitudes.

For all optimizers, the momentum decay parameter is set to $\beta_1 = 0.9$, the variance decay parameter is set to $\beta_2 = 0.999$, and the learning rate multiplier is found by grid search on the Train optimizee over a grid of $\{5 \times 10^{-4}, 1 \times 10^{-3}, 2 \times 10^{-3}, \dots 1\}$.

### C.3 ADDITIONAL TRAINING DETAILS

During amalgamation, we apply a number of techniques from previous Learning to Optimize literature in order to boost training:

- *Curriculum Learning*: We apply curriculum learning (Chen et al., 2020a) to progressively increase the unrolling steps across a maximum of 4 stages with length 100, 200, 500, and 1000. During curriculum learning, checkpoints are saved and validated every 40 "meta-epochs," which refers to a single optimizee trajectory trained with truncated back-propagation.

- *Random Scaling*: We apply random scaling (Lv et al., 2017) to reduce overfitting to the gradient magnitudes of the training problem. This random scaling is only applied to the amalgamation target; amalgamation teachers receive "clean" (unscaled) gradients.

- *Warmup*: Instead of initializing each training optimizee with random weights, we first apply 100 steps of SGD optimization as a "warmup" to avoid the turbulent initial phase of optimizing neural networks. A SGD learning rate of 0.01 is used during this period, and was chosen to be very conservative on all optimizees tested.

These techniques are also applied to all of our baselines, except for Random Scaling is only applied to baselines using the RNNProp architecture since we find that it harms the performance of other optimizer architectures.

## C.4 TRAINING COST

Table 4 provides a summary of the training costs for each amalgamation method and baseline are provided. For optimal choice amalgamation, this includes both training the optimal choice optimizer and amalgamation training. All values are reported as the mean across 8 replicates.

Table 4: Amalgamation and baseline training times.

| Method | Time (hours) | Method | Training Time (hours) |
|---|---|---|---|
| Mean | 3.86 | Random | 5.27 |
| Min-max | 3.85 | Adversarial | 10.53 |
| Choice (small) | 5.28 | RNNProp | 2.39 |
| Choice (large) | 6.35 | Stronger Baselines | 3.56 |

All experiments were run on single nodes with 4x Nvidia 1080ti GPUs, providing us with a meta-batch size of 4 simultaneous optimizations. In order to replicate our results, GPUs with at least 11GB of memory are required, though less memory can be used if the truncation length for truncated back-propagation is reduced.

## D STABILITY DEFINITIONS

In this section, we provide the mathematical definition and measurement details of meta-stability, evaluation stability, and optimization stability.

### D.1 META-STABILITY AND EVALUATION STABILITY

In order to quantify meta-stability and evaluation stability, we first summarize the performance of each evaluation using the best validation loss obtained and the training loss of the last epoch. Then, we model the best validation loss $Y_{ij}^{(\text{val})}$ and final training loss $Y_{ij}^{(\text{train})}$ for replicate $i$ and evaluation $j$ with the linear mixed effect model

$$Y_{ij} = \mu + \alpha_i + \varepsilon_{ij}, \tag{9}$$

where $\mu$ is the true mean, $\alpha_i$ are IID random variables representing the meta-stability of the amalgamated optimizer, and $\varepsilon_{ij}$ are IID random variables representing the evaluation stability of each replicate. The meta-stability and evaluation stability are then quantified by standard deviations $\sigma_\alpha$ and $\sigma_\varepsilon$.

### D.2 OPTIMIZATION STABILITY

To measure optimization stability, we model the validation loss $L_{ij}(t)$ at epoch $t$ for replicate $i$ and evaluation $j$ as

$$L_{ij}(t) = \beta_{ij}(t) + \eta_{ij}^{(t)} \tag{10}$$

for smooth function $\beta_{ij}(t)$ which represents the behavior of the evaluation and random variable $\eta_{ij}^{(t)}$ which captures the optimization stability; we assume that $\eta_{ij}^{(t)}$ is IID with respect to $t$.

In order to estimate $\sigma_\eta$, we first estimate $\beta_{ij}(t)$ by applying a Gaussian filter with standard deviation $\sigma = 2$ (epochs) and filter edge mode "nearest," and $\hat{\sigma}_\eta^{(ij)}$ is calculated accordingly. Finally, $\sigma_\eta^{(ij)}$ is treated as a summary statistic for each evaluation, and the mixed effect model described previously (Equation 9) is fit to obtain a final confidence interval for the mean optimization stability.

# E  ADDITIONAL RESULTS

## E.1  EVALUATION STABILITY

Table 5 summarizes the evaluation stability of analytical and amalgamated optimizers. All optimizers obtain similar evaluation stability, except for cases where an optimizer cannot reliably train the optimizee at all such as Momentum, AddSign, and PowerSign on the Deeper CNN. In these cases, the optimizer consistently learns a constant or random classifier, which results in very low variance and high "stability."

Table 5: Evaluation stability of analytical and amalgamated optimizers; all optimizers are amalgamated from the small pool, except for Optimal Choice Amalgamation on the large pool, which is abbreviated as "large". A dash indicates optimizer-problem pairs where optimization diverged.

| | | | | | Best Log Validation Loss | | | | | |
| Problem | Adam | RMSProp | SGD | Momentum | AddSign | PowerSign | Mean | Min-Max | Choice | Large |
|---|---|---|---|---|---|---|---|---|---|---|
| Train | 0.068 | 0.048 | 0.048 | 0.068 | 0.068 | 0.049 | 0.064 | 0.072 | 0.064 | 0.058 |
| MLP | 0.044 | 0.046 | 0.027 | 0.045 | 0.022 | 0.039 | 0.052 | 0.048 | 0.046 | 0.042 |
| Wider | 0.060 | 0.123 | 0.046 | 0.056 | 0.072 | 0.059 | 0.071 | 0.064 | 0.063 | 0.055 |
| Deeper | 0.084 | 0.075 | 0.047 | 1.971 | 1.679 | — | 0.085 | 0.062 | 0.100 | 0.087 |
| FMNIST | 0.011 | 0.017 | 0.015 | 0.000 | 0.019 | 0.017 | 0.023 | 0.020 | 0.018 | 0.021 |
| SVHN | 0.099 | 0.049 | 0.019 | 0.089 | 0.037 | 0.026 | 0.095 | 0.360 | 0.064 | 0.081 |
| CIFAR-10 | 0.049 | 0.025 | 0.030 | 0.000 | 0.031 | 0.032 | 0.040 | 0.044 | 0.031 | 0.021 |
| ResNet | 0.042 | 0.054 | — | — | — | — | 0.072 | 0.052 | 0.044 | 0.040 |
| Small Batch | 0.047 | 0.119 | 0.079 | 0.068 | 0.056 | 0.106 | 0.085 | 0.078 | 0.065 | 0.064 |
| MNIST-DP | 0.065 | 0.055 | 0.154 | 0.151 | 0.269 | 0.229 | 0.075 | 0.076 | 0.070 | 0.069 |

## E.2  LARGER EVALUATIONS

In order to explore the limits of our optimizer, we evaluated the amalgamated optimizer with a 52-layer ResNet (763882 parameters — 40x the train network size), and the same 52-layer ResNet on CIFAR-100 instead of CIFAR-10. These results are compared to CIFAR-10 on a 2-layer network and CIFAR-10 on a 28-layer ResNet using Adam as a single baseline (Figure 8).

While our amalgamated optimizer has significant performance advantages on the shallow CIFAR-10 network in our original evaluations and achieves performance parity in the 28-layer ResNet, the amalgamated optimizer can no longer perform as well as the oracle once we reach 52 layers and change to CIFAR-100.

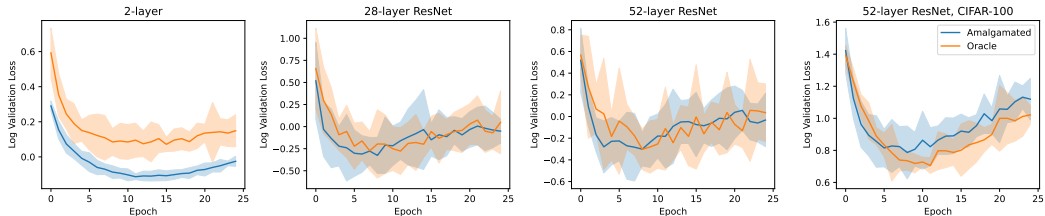

Figure 8: Amalgamated optimizer ablations on problems of increasing size relative to the training problem.

## E.3  ADDITIONAL PLOTS

In this section, we include plots providing alternate versions of Figures 2, 3, and 5 in the main text which had some outliers cropped out in order to improve readability.

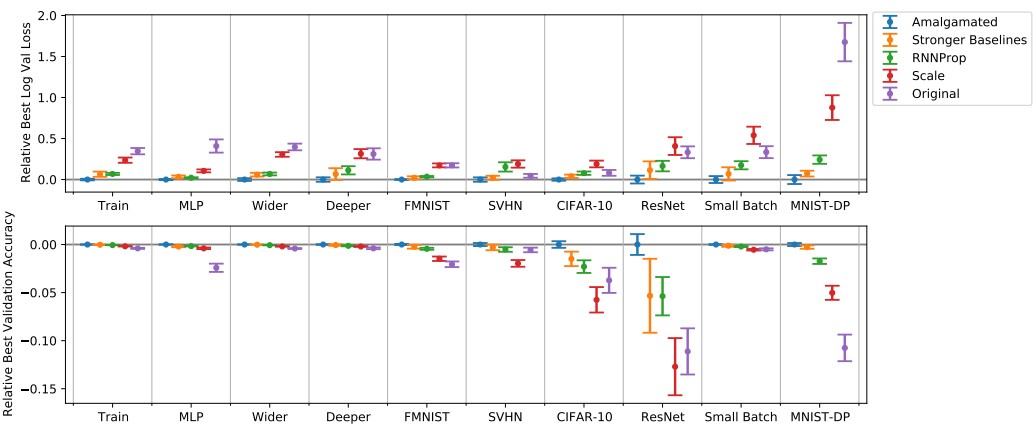

Figure 9: Uncropped version of Figure 2. Poor performance of "Scale" and "Original" on small batch and differentially private training cause differences between the best performers (Amalgamated and Stronger Baselines) to be unreadable. A version showing the best validation accuracy is also included (higher is better); the accuracy results largely preserve relative differences between methods, and lead to the same conclusion.

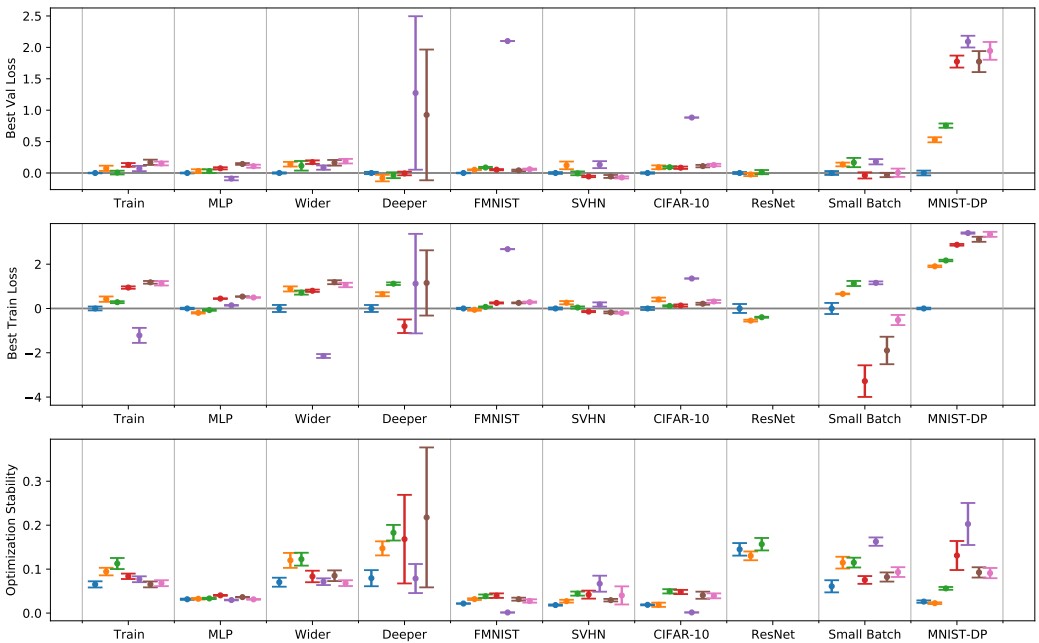

Figure 10: Uncropped version of Figure 5 including similar plots for Training and Validation loss.

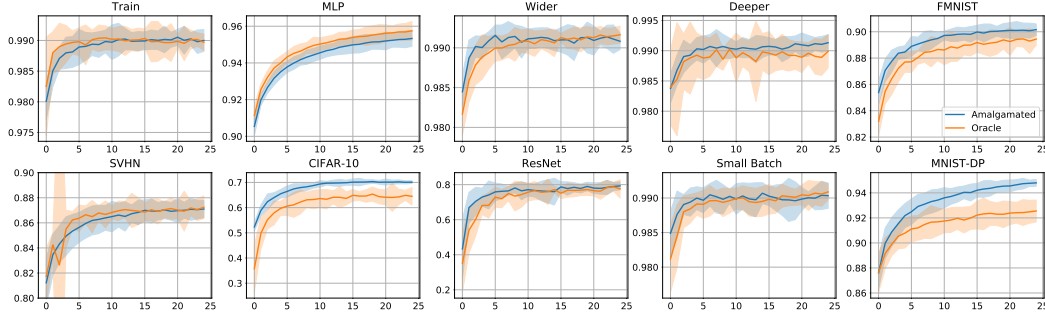

Figure 11: An accuracy version of Figure 3 comparing the best Amalgamated Optimizer (blue) and the Oracle Optimizer (orange); the shaded area shows $\pm 2$ standard deviations from the mean. The title of each plot corresponds to an optimizee; full definitions can be found in Appendix B. The amalgamated optimizer performs similarly or better than the Oracle analytical optimizer on problems spanning a variety of training settings, architectures, and datasets, and has the largest advantage on more difficult problems such as CIFAR-10, ResNet, and MNIST-DP.

### E.4 INPUT PERTURBATION

When applying input perturbations, we perturb the inputs to the optimizer, or the optimizee gradients, instead of the optimizer weights:

$$\theta_{i+1} = \theta_i - P(\nabla_{\theta_i} \mathcal{M}(\boldsymbol{x}_i, \theta_i) + \mathcal{N}(0, \sigma^2 I)). \tag{11}$$

We tested magnitudes $\sigma = 10^{-1}$ and $\sigma = 10^{-2}$ on a smaller experiment size of 6 replicates using Choice amalgamation on the small pool as a baseline; these results are given in Table 6. Many experiment variants remain relating to input noise such as adding noise proportional to parameter norm or gradient norm or trying smaller magnitudes of noise, and this may be a potential area of future study. However, we believe that input noise is generally not helpful to optimizer amalgamation, and did not study it further.

Table 6: Meta-Stability with varying magnitudes of Input Perturbation

| Magnitude | Meta-stability |
|---|---|
| $\sigma = 0$ | 0.104 |
| $\sigma = 10^{-2}$ | 0.485 |
| $\sigma = 10^{-1}$ | 1.637 |

### E.5 BASELINES WITH RANDOM PERTURBATION

Our perturbation methods can be applied to any gradient-based optimizer meta-training method, including all of our baselines. To demonstrate this application, we trained 8 RNNProp replicates with Gaussian perturbations with magnitude $1 \times 10^{-4}$; all other settings were identical to the RNNProp baseline. With perturbations, the RNNProp baseline is significantly improved, though not enough to match the performance of our amalgamation method.

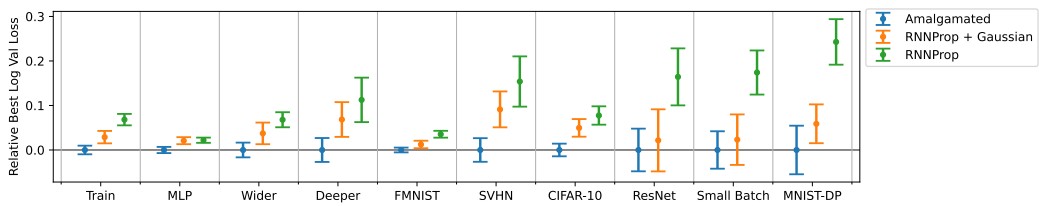

Figure 12: Comparison of 8 replicates amalgamated from the Large pool using Choice amalgamation with RNNProp baseline with Gaussian perturbations, with magnitude $1 \times 10^{-4}$. With perturbations, the RNNProp baseline is significantly improved, though not enough to match the performance of our amalgamation method.

