# OpenReview forum: "Optimizer Amalgamation"
_ICLR.cc/2022/Conference — ICLR 2022 Poster_

### Official Review · Reviewer_kuqL · 2021-10-26

**Correctness:** 3
**Technical Novelty And Significance:** 2
**Empirical Novelty And Significance:** 2
**Recommendation:** 6
**Confidence:** 4

**Main Review:**

Strengths:
- This paper is mostly clear and organized, except for some points mentioned in Weaknesses below.
- The paper made an interesting attempt to distill the knowledge from analytical optimizers with three amalgamation methods and conducted comprehensive experiments to show the effectiveness of the amalgamated optimizer.


Weaknesses:
- There are some places that the paper did not describe the content clearly, especially for the definition of stability. When to use $Y_{ij}^\text{val}$ and $Y_{ij}^\text{train}$ in Eq. (9) was not discussed and thus it is confusing to me how to derive the meta-stability and evaluation stability.
- In terms of both the validation loss and the validation accuracy, the amalgamated optimizer did not have obvious advantages over the oracle optimizer, as shown in Figure 4 and Figure 9, which makes the method not very practical.
- Although the authors claimed that they tried to distill the knowledge from traditional optimizers, it is also related to imitation learning. However, there was no discussion about the different between knowledge distillation and imitation learning when trying to learn the optimization pattern from analytical optimizers. Some imitation baselines might be incorporated into the meta-loss as well.
- I am wondering whether first initializing the learned optimizer via the proposed amalgamation methods or imitation learning methods and then training it with the meta loss will have better performance. The reason is that randomly initialized optimizer is hard to train at the beginning and might disturb the whole training trajectory.
- It seems that adversarial perturbation performed similarly as random perturbation. Since adversarial perturbation requires more computing resources, then why not just use random perturbation? Are there any other advantages of adversarial perturbation?
- I am also interested in the performance of the choice optimizer C. Maybe the authors can expand on it, such as comparing C with the analytical optimizers it learns from.
- I don't quite understand the choice of l2-log loss. Is there any related paper using this type of distillation loss?  Maybe the authors can try the gradient matching loss in [1], which is shown to be effective for dataset distillation and works with a similar number of parameters.
- The time and memory cost for training the amalgamated optimizer is not reported, which is important for the practicality of the method.

[1] Zhao, B., Mopuri, K. R., & Bilen, H. (2020). Dataset condensation with gradient matching. https://openreview.net/forum?id=mSAKhLYLSsl


**Summary Of The Paper:**

This paper proposed a new problem called Optimizer Amalgamation and made an attempt to obtain a more powerful learned optimizer from several analytical optimizers. More concretely, three amalgamation losses are designed to train the amalgamated optimizer. In the meanwhile, two types of noise, random gaussian perturbation and projected gradient descent are incorporated into the training objective to increase the stability of the optimizer. The evaluation part compared the amalgamated optimizer with its original teachers, i.e., those analytical optimizers, and some L2O baselines on image classification tasks.

**Summary Of The Review:**

Generally, I think the paper is marginally below the acceptance threshold. Although optimzer amalgamation is interesting, it seems expensive to train such an optimizer based on my experience and the authors did not report any time/memory complexity. Besides, the performance is not improved obviously compared with the best analytical optimizer, and the paper lacks the detailed discussion about imitation learning.

---

> ### Author Response · Authors · 2021-11-15
> **Response to Reviewer kuqL (1/2)**
>
> Thank you for your detailed response; your evaluation drives us to improve our paper with more details and experiments. Our responses to your concerns are provided below.
>
> **[Q1 Meta-Stability and Evaluation Stability]**
> Since the definition of meta-stability and evaluation stability are somewhat lengthy, we included them in Appendix D in order to make the page limit.
>
> **[Q2 Oracle Optimizer Comparison]**
> While our method is not convincingly better than the oracle optimizer, practitioners do not have access to the oracle: the only way to obtain the oracle optimizer is to try all analytical optimizers on the problem and pick the best one. The amalgamated optimizer has clear advantages over choosing a random analytical optimizer or using the same analytical optimizer for all problems. This can be seen in figure 5: no single analytical optimizer always performs well, and a randomly chosen analytical optimizer will most likely perform significantly worse than the amalgamated optimizer.
>
> **[Q3 Distillation vs Imitation]**
> In general, Knowledge Distillation refers to imitating a model, while Imitation Learning refers to imitating some global policy. However, in our context, the optimizer output can be regarded as both the output of a model and the output of a policy, so Imitation Learning and Knowledge Distillation are the same thing here. This means that we can also view Imitation Learning as a special case of Optimizer Amalgamation where we amalgamate from a single optimizer, and all amalgamation methods collapse to a constant. We have added this discussion to our paper in section 3.2.
>
> In our baselines, we include the “Stronger Baselines” baseline (Chen et al., 2020a) which uses imitation learning. If there are any other imitation learning baselines for learned optimizers that you are aware of, we can add them to our paper.
>
> **[Q4 Initialization with Amalgamation]**
> We trained a single replicate using optimal choice amalgamation and the scheme you describe. The replicate trained using initialization by amalgamation performs similarly to optimal choice amalgamation; more conclusive results will require a full evaluation.
>
> | Value | Amalgamation throughout Training | Initialization by Amalgamation |
> | --- | --- | --- |
> | Best Log Val Loss | -3.49 | -3.53 |
> | Final Training Loss | -5.58 | -5.61 |
>
> **[Q5 Adversarial vs Random]**
> Our goal is to explore both random and adversarial perturbations in our new problem domain. As you observed, we find that they perform similarly empirically (figure 6), and we agree that random perturbations are more practical. We have made this recommendation explicit in our paper.
>
> **[Q6 Choice Optimizer Baseline]**
> We evaluated the choice optimizer for the large pool on our 10 comparison problems; these results have been added to figure 4.
>
> Overall, the choice optimizer performs similarly to the amalgamated optimizer and the oracle optimizer, except for CIFAR-10, where the amalgamated optimizer is better (though the choice optimizer does still beat the oracle), small batch training, where the choice optimizer overfits very aggressively, and differentially private training, where the choice optimizer diverges after one epoch of training.
>
> **[Q7 L2 Log Loss]**
> Log losses are commonly used in training learned optimizers. For example, Wichrowska et al. (2017) use a log meta-loss. Our imitation learning baseline, Chen et al. (2020a), also uses a log objective in its implementation.
>
> Since we used these two implementations as a reference for our meta-training framework, and we use many of the same hyperparameters, we continue to use a log objective. Many other works do not use a log objective, and our amalgamation process would probably work with the hyperparameters re-tuned for a non-log objective.
>
> The gradient matching loss you raise seems promising; along with l2 loss, log l2-loss and other unroll loss weightings (applying a non-uniform weight to losses across an unroll), the choice of what loss function to use is an open question that we will explore in the future.
>
> (continued)

---

> ### Author Response · Authors · 2021-11-15
> **Response to Reviewer kuqL (2/2)**
>
> **[Q8 Time and Memory Cost]**
> On average, amalgamating an optimizer using Optimal Choice amalgamation on the large pool takes a total of 6.35 hours, including training the choice optimizer. Our experiments are run on machines with 4x Nvidia 1080ti GPUs. We have added this cost to section 5 and a full accounting of our training costs to Appendix C.4.
>
> While the amalgamation process is quite expensive, our vision is that amalgamated optimizers (or more generally, learned optimizers) will be distributed along with pre-trained or meta-learned weights as part of a larger meta-learning ecosystem. These optimizers would be meta-trained to fine tune some or all of those parameters for specific tasks.
>
> This training and distribution process would be done similarly to how pre-trained weights for large convolutional backbones or large transformers are currently distributed: trained using massive amounts of data (many training problems) and compute (larger training problems, more and longer unrolls) by a few large institutions at great cost, but widely reused by practitioners of all capabilities and sizes.

---

> ### Author Response · Authors · 2021-11-23
> **Response to Reviewer kuqL**
>
> Dear Reviewer kuqL,
>
> Thank you for your suggestions; your feedback has led to several changes and new experiments, improving our paper. In response to your suggestions, we have updated our paper to include additional discussion on imitation learning, made explicit our recommendations for random perturbations, and added an accounting of our training costs. We have also run new experiments, including evaluations of the choice optimizer as a new baseline.
>
> We would greatly appreciate if you could check our response to see if our response satisfies your concerns, and hope to engage in further discussion to answer any future questions you may have.
>
> Best wishes,
> Optimizer Amalgamation Authors

---

> ### Author Response · Authors · 2021-11-26
> **Request for Comment**
>
> Dear Reviewer kuqL,
>
> We sincerely hope to have further discussion with reviewer kuqL to see if our response resolves their concerns. We are confident that our response should clear any confusion; we can clarify more if there is more need. We are happy to answer any additional questions and provide more information.
>
> We genuinely hope reviewer kuqL could kindly check our response. Thank you!
>
> Best wishes,
>
> Authors

---

> ### Author Response · Authors · 2021-11-27
> **Request for Comment**
>
> Dear Reviewer kuqL,
>
> There are less than 48 hours remaining in the discussion period, and we would really like to have further discussion regarding your concerns and our responses. We would greatly appreciate if you could reply to our response.
>
> Thanks,
> Authors

---

> > ### Comment · Reviewer_kuqL · 2021-11-28
> > **Thanks for the response**
> >
> > Thanks for your response. It solved my concerns and I will increase my score to 6.

---

### Official Review · Reviewer_h9GD · 2021-10-30

**Correctness:** 3
**Technical Novelty And Significance:** 3
**Empirical Novelty And Significance:** 3
**Recommendation:** 6
**Confidence:** 3

**Main Review:**

Strengths
=

1. The paper is generally well-written and easy to follow.

2. The proposed optimizer amalgamation method is interesting and effective.

3. The experiment is comprehensive and supports the main claim of the paper.



Weakness
=
Selecting an appropriate optimizer for a given problem is not a new research topic, and many papers exist in the areas of algorithm selection, algorithm portfolio, and meta-learning. I think one weakness of the paper is that it lacks the discussion about the related work, and how the paper can be better positioned in the literature.


**Summary Of The Paper:**

This paper presents a new optimizer amalgamation method to combine a pool of optimizers into one in order to achieve stronger problem-specific performance. Three differentiable amalgamation mechanisms are designed and stabilization methods are explored. The proposed method is empirically shown to be effective when compared to a large number of baselines.



**Summary Of The Review:**

Although the research topic is old, the proposed optimizer amalgamation method looks interesting and the experimental results verified the effectiveness of the proposed method.

---

> ### Author Response · Authors · 2021-11-15
> **Response to Reviewer h9GD**
>
> Thank you for your feedback. We have updated our related work by adding a section, “Other Approaches,” describing other approaches to solving the “big problem,” AutoML, of how to better select models and train them:
> - We first highlight meta-learning approaches that tackle the initialization of neural networks instead of training such as MAML (Finnet al., 2017) and Reptile (Nichol et al., 2018).
> - Furthermore, we include hyperparameter optimization methods which, like our work, focus on the training process such as hypergradient descent (Baydin et al., 2017) and bayesian hyperparameter optimization (Snoek et al., 2012).
> - Finally, we highlight approaches that also use a “portfolio” of algorithms and select some subset applied to problem domains such as Linear Programming (Leyton-Brown et al., 2013) and SAT solvers (Xu et al., 2008).

---

### Official Review · Reviewer_ZiUc · 2021-11-02

**Correctness:** 3
**Technical Novelty And Significance:** 3
**Empirical Novelty And Significance:** 3
**Recommendation:** 6
**Confidence:** 4

**Main Review:**

Strengths
* The problem is well motivated as a solution to choosing from the large pool of possible optimizers. It naturally follows from prior work on knowledge distillation and amalgamation to instead distill from multiple optimizers.
* Empirically, optimizer amalgamation performs better (and sometimes significantly so) than baselines from learning to optimize. It also performs favorably to the best choice of analytic optimizer.
* Paper is clearly written and easy to follow.

Weaknesses
* It is a bit unclear how well an amalgamated optimizer can generalize to various settings -- do we need to amalgamate for each specific problem?
* I have some concerns with the experiments discussed in the questions below.
* Overall, the experiments are conducted in relatively small settings. It would be useful to know whether an optimizer learned with amalgamation could be used in a large experiments. Such experiments do not need to be extensive, but would improve the significance of the paper.

Questions
1. It would be interesting to explore further the choice of optimizer pool and its effects on performance. This can be done by choosing subsets of the six optimizers you consider. SGD and Adam would be especially interesting given their popularity.
2. In the case of mean amalgamation, do you observe the phenomena of the optimizer sticking to one of the teachers?
3. Can the weight space perturbations be applied to the baselines to improve their stability?

Minor
* reference under knowledge distillation is undefined

**Summary Of The Paper:**

This paper discusses the problem of selecting a neural network optimizer from a pool of possible optimizers. They propose three variants of a meta-algorithm which combines optimizers from the pool, whereby a differentiable meta-loss is defined on the training loss achieved by selection protocol. They show that their algorithm, equipped with weight space training noise leads to better performance on a variety of problems.

**Summary Of The Review:**

I think this paper is marginally above the acceptance threshold. The algorithm proposed is well motivated for a specific task, and the authors conduct experiments with various ablations to their methods. I have some concerns listed in the questions, but overall believe this to be a good first step in tackling the problem.

---

> ### Author Response · Authors · 2021-11-15
> **Response to Reviewer RiUc**
>
> Thank you for your suggestions. We have revised our paper based on your feedback, and include responses to your questions below.
>
> **[W1: Generalization]**
> We do not need to re-amalgamate our optimizer for each new problem; for our experiments, we state that all optimizers were trained on the Train ConvNet using MNIST. Our experiments then demonstrate generalization to a range of different datasets (FMNIST, SVHN, CIFAR), different architectures (MLP, wider, deeper, ResNet), and different training settings (small batch, differential privacy).
>
> **[W2: Large-Scale Experiments]**
> The ResNet architecture used in our evaluation on CIFAR-10 has 372330 parameters, which is >20x larger than the Training network on MNIST, which has only 18122 parameters. We have added the parameter count of each problem to the summary table in Appendix B to provide an at-a-glance overview of problem size.
>
> Note that since the experiments included in our paper require amalgamating 96 optimizers (12 settings x 8 replicates), we chose a very small training problem. As such, even the 28-layer ResNet we use, which is otherwise small compared to 152-layer ResNets and Wide ResNets, is massive compared to the training problem.
>
> To further explore the behavior of our amalgamated optimizer at scale, we have run a small-scale (5 runs; single replicate; Adam as single baseline) evaluations of our choice network trained on the large pool on larger problems to test it “to failure.”
>
> Currently, we have evaluated (1) a deeper ResNet with 52 layers (763882 parameters; 40x Train network) and (2) the same 52-layer ResNet on CIFAR-100 instead of CIFAR-10. For the time being, these results are provided in Appendix E.2.
>
> Based on these preliminary evaluations, we observe that while our amalgamated optimizer has significant performance advantages on the shallow CIFAR-10 network in our original evaluations and achieves performance parity in the 28-layer ResNet, the amalgamated optimizer can no longer perform as well as the oracle once we reach 52 layers and increase the dataset complexity to CIFAR-100.
>
> We will follow up with larger-scale complete comparisons, which we will add to our paper once complete.
>
> We will follow up with larger-scale complete comparisons, which we will add to our paper once complete.
>
> **[Q1: Optimizer Pool Subsets]**
> Our experiments currently include a smaller subset of Adam and RMSProp forming the Small optimizer pool. We ran a single new amalgamation experiment using Adam and SGD for the pool as suggested, and compare it to the results with Adam and RMSProp.
>
> | Value | (Adam, RMSProp) | (Adam, SGD) |
> | --- | --- | --- |
> | Best Log Val Loss | -3.49 | -3.35 |
> | Final Training Loss | -5.58 | -3.89 |
>
> While the optimizer amalgamated using Adam and SGD performs significantly worse, this may be because the hyperparameters need further tuning or because we amalgamated only a single replicate and are simply unlucky. Since performing the full evaluations necessary to draw a statistically significant conclusion is extremely expensive, we will take this as our future work.
>
> **[Q2: Mean Amalgamation Teacher Sticking]**
> Our explanation of sticking is only intended as motivation and justification as to why we believe mean amalgamation is not sufficient. We believe that it is very difficult, if not impossible, to measure this type of sticking since it should occur on a per-iteration and per-direction basis in the parameter space. The effect is further obscured in our large-scale experiments by the contribution of meta-loss, stochasticity, and parallelism (running “meta-batches” of 4 unrolls simultaneously).
>
> We make an attempt to measure this effect, which we have added to Appendix E.5. We ran a small-scale experiment training 50 unrolls with unroll length 100 using only amalgamation loss and without any parallel meta-training, recorded the distillation loss to each teacher at the end of the unroll, and plotted this over time. From our experiment, we observe that Adam is slightly favored over RMSProp overall; however, we are not convinced that this truly demonstrates the sticking effect due to the reasons described above.
>
> **[Q3: Perturbations Applied to Baselines]**
> We will add additional experiments evaluating the strongest baselines (specifically Stronger Baselines and RNNProp) with perturbations.
>
> **[Undefined Reference]**
> Thank you for pointing this out; we have fixed this typo in our references.

---

> > ### Comment · Reviewer_ZiUc · 2021-11-29
> > **Thank you for the response**
> >
> > I have read the other reviews and author's responses. I appreciate the more detailed evaluation in Appendix E2. However, the result raises some concerns about the ability of the method to scale to larger problems. I think CIFAR is still relatively small compared to other problem settings tackled in deep learning today.
> >
> > I realized in revisiting the paper that learning rate schedules do not seem to be discussed, and I believe that they are relevant in the context of amalgamating optimizers. Specifically, we might expect the "choice" amalgamation to behave differently under different learning rate regimes. It would be informative to read some discussion in the next revision.
> >
> > Overall, the other responses are satisfactory and despite some qualms about scalability to larger problems, I will maintain my score.

---

> > > ### Author Response · Authors · 2021-11-30
> > > **Response to Reviewer ZiUc**
> > >
> > > Many thanks for your further constructive feedback. We try our best to deliver some extra analysis and experiments before the deadline; more thorough results and analyses will be added in the final version, which we will update when the portal reopens.
> > >
> > > We trained optimal choice optimizers using the small pool (Adam and RMSProp) using a fixed Adam learning rate of 5e-3 and varying RMSProp learning rates, and record the average weight of the RMSProp choice across a single evaluation; these results are given in the table below.
> > >
> > > | Adam LR | RMSProp LR | RMSProp Proportion |
> > > | --- | --- | --- |
> > > | 5e-3 | 1e-4 | 0.609 |
> > > | 5e-3 | 2e-4 | 0.762 |
> > > | 5e-3 | 5e-4 | 0.787 |
> > > | 5e-3 | 1e-3 | 0.690 |
> > > | 5e-3 | 2e-3 | 0.438 |
> > > | 5e-3 | 5e-3 | 0.111 |
> > > | 5e-3 | 1e-2 | 0.083 |
> > >
> > > As the RMSProp learning rate increases, the weight given to RMSProp increases as the learning rate becomes better tuned to the problem. When the learning rate of RMSProp eventually becomes too large, the weight begins to decrease, eventually becoming near zero.

---

### Official Review · Reviewer_Zbiy · 2021-11-07

**Correctness:** 3
**Technical Novelty And Significance:** 3
**Empirical Novelty And Significance:** 3
**Recommendation:** 6
**Confidence:** 4

**Main Review:**

Strength:

1. The idea of distilling from multiple "teacher" optimizer is novel although distilling from one "teacher" optimizer has been proposed before as imitation learning in L2O.

2. The three mechanisms proposed to distill from several "teachers" are interesting, especially the optimal choice amalgamation, which could shed light some further improvements for L2O.

3. The use of random and adversarial pertubation on weights to improve robustness is a nice technique to help with the stability that is a big issue in L2O.

Weakness:

1. The comparison of the amalgamated optimizer with the analytical optimizers are interesting, but the advantage of the analytical optimizers is that they tend to work reasonably well for a wide range of problem or model sizes. Since the amalgamated optimizer is still learned on specific settings, I wonder how far it could generalize and when would such generalization break? For example, if the model size or the training steps is 10X or 100X larger, would the learned optimizer still perform well? These could shed some light on the limitation of current approaches and inspire future works.

2. The "oracle optimizer" is using the best validation loss among the analytical optimizer's in the pool, but a stronger oracle might be the optimal choice optimizer that uses the trained LSTM to pick which optimizer to use at each step. It would help to add this stronger baseline into the analysis and comparison in the experiments since it can help understand whether the bottleneck is in the optimal choice LSTM or the distillation process.

3. It might help to add more details about the Optimal Choice Amalgamation to help with reproducibility. For example, "the LSTM takes the outputs of each optimizer in the pool, the layer type, and time step as inputs", I wish there are more descriptions (maybe in the supplementary material) about the details such as how the inputs are encoded and how often are the LSTM updated to help with reproducibility. It would also be very helpful if the code can open sourced.

4. It would be helpful to add some experiments to justify that it is necessary to distill from multiple optimizers instead of just one since the "small" setting just uses two optimizers anyway and the gain of the proposed approach from baseline might come more from the better stability in training due to perturbation.


**Summary Of The Paper:**

This work presents an approach to distill several "teacher" optimizer into a "student" optimizer through learning to optimize (L2O). They compared three different approaches for doing such distillation, proposed to use random and adversarial perturbation to help with stability, and compared with the analytical optimizers that it distills from and the L2O baselines and showed superior performance.


**Summary Of The Review:**

This work presents a novel approach called optimizer amalgamation that distills multiple "teacher optimizer" into a learned "student" optimizer. They compared three different approaches and proposed to use perturbation to improve stability in L2O. The result is promising compared to previous baselines, but it would be helpful if more analyses (see weakness section) can be done to understand the importance and necessity of different components and more details of the proposed approach can be released to help with reproducibility.

---

> ### Author Response · Authors · 2021-11-15
> **Response to Reviewer Zbiy**
>
> Thank you for your feedback. Your questions raise a number of areas where we can improve our paper; we include a response to your questions below.
>
> **[Q1 Generalization to larger models]** The ResNet architecture used in our evaluation on CIFAR-10 has 372330 parameters, which is >20x larger than the Training network on MNIST, which has only 18122 parameters. We have added the parameter count of each problem to the summary table in Appendix B to provide an at-a-glance overview of problem size.
>
> Note that since the experiments included in our paper require amalgamating 96 optimizers (12 settings x 8 replicates), we chose a very small training problem. As such, even the 28-layer ResNet we use, which is otherwise small compared to 152-layer ResNets and Wide ResNets, is massive compared to the training problem.
>
> You raise an excellent idea of testing our optimizer “to failure” to see the limits of our method. We have run a small-scale (5 runs instead of 10; single replicate) evaluation of our choice network trained on the large pool on (1) a deeper ResNet with 52 layers (763882 parameters; 40x Train network) and (2) the same 52-layer ResNet on CIFAR-100 instead of CIFAR-10; we used Adam as a single baseline since it is the Oracle Optimizer for the base ResNet problem.
>
> For the time being, these results are provided in Appendix E.2. Based on these preliminary evaluations, we observe that while our amalgamated optimizer has significant performance advantages on the shallow CIFAR-10 network in our original evaluations and achieves performance parity in the 28-layer ResNet, the amalgamated optimizer can no longer perform as well as the oracle once we reach 52 layers and increase the dataset complexity to CIFAR-100.
>
> We will follow up with larger-scale complete comparisons, which we will add to our paper once complete.
>
> **[Q2 Choice Optimizer Baseline]**
> We evaluated the choice optimizer for the large pool on our 10 comparison problems; these results have been added to figure 4.
>
> Overall, the choice optimizer performs similarly to the amalgamated optimizer and the oracle optimizer, except for CIFAR-10, where the amalgamated optimizer is better (though the choice optimizer does still beat the oracle), small batch training, where the choice optimizer overfits very aggressively, and differentially private training, where the choice optimizer diverges after one epoch of training.
>
> **[Q3 Architecture Details and Open Source]** We have added a detailed description of the RNNProp and Optimal Choice optimizer architectures to Appendix C.
>
> We plan to make our code open source after publication, and have put significant effort into making our code readable, modular, and easily modifiable. If you would like to take a look, our code is currently included in the supplemental with some details and scripts censored for anonymization; we have added an open-source statement to our abstract, which will link to a GitHub repository after publication.
>
> **[Q4 Distillation with One Optimizer]** When distilling from a single optimizer, all three methods of distillation become equivalent to imitation learning to a single optimizer, which is the Stronger Baselines comparison baseline; we will work on additional experiments evaluating Stronger Baselines and RNNProp with perturbations.

---

### Author Response · Authors · 2021-11-15
**Revision Summary**

We would like to thank all reviewers for providing useful feedback for our paper. A summary of the additions and corrections to our paper can be found below.

**Additions**

- Added an extra related work section discussing the broader AutoML space.
- Added explicit recommendation for practitioners to use random perturbations in section 5.2.
- Added a detailed description of the RNNProp and Optimal Choice optimizer architectures to Appendix C.
- Added an accounting of our training costs in Appendix C.4 with summary statement in section 5.
- Inserted Appendix E.2 containing evaluations on larger problems (52-layer ResNet and CIFAR-100). These results are incomplete and will be updated.
- Added Appendix E.5 containing a short mean amalgamation sticking experiment.

**Modifications**

- Reworded section 3.2, “Distillation Loss” to explicitly describe the relationship between knowledge distillation and imitation learning.
- Updated Figure 4 (and the description) to include the choice optimizer as a baseline.
- Modified the summary table in Appendix B to include the parameter count of each problem, providing an at-a-glance overview of problem size.

**Corrections**

- Corrected a missing citation of “Learned Optimizers that Scale and Generalize” (Wichrowska et al) for the Scale baseline in the experiment section.
- Fixed a special character in an author name not being understood by BibTex and turning into “undefined”.

We hope our responses and preliminary experiments provided can provide additional insight into our work. We are currently working on other additional experiments, and will update our paper and post replies here when they are complete.

---

### Author Response · Authors · 2021-11-22
**Second Revision with Completed Experiments**

We have modified our submission to include updated experiment results with larger scale evaluations:

**Evaluation of larger problems**: we completed evaluation of a full 10 runs of the ResNet-52 and ResNet-52 / CIFAR-100 evaluation problems. Appendix E.2, figure 8 has been updated with the full results, which are more or less identical to the previous results, leaving our previous conclusions unchanged.

**Evaluation of baselines with perturbations**: Appendix E.5 has been added with an experiment comparing Choice amalgamation from the large pool, our RNNProp baseline, and RNNProp with Gaussian perturbations using magnitude selected from our ablations in Figure 6. With Gaussian perturbations, the RNNProp baseline is significantly improved, though not enough to match the performance of our amalgamation method.

---

### Decision · Program_Chairs · 2022-01-20

**Decision:**

Accept (Poster)

**Comment:**

This paper introduces a meta-learning approach to "amalgamate" optimizers. The reviewers all found the idea interesting and unanimously found it to be acceptable for publication. In particular, I appreciate that the authors expanded their results to include more larger problems. One of the outstanding questions that would be interesting to address in future work is the use of tuned learning rate schedules.